

# Plant community recovery from intense deer grazing depends on reduction of graminoids and the time after exclosure installation in a semi-natural grassland

Chiaki Otsu[1], Hayato Iijima[2] and Takuo Nagaike[1]

[1] Department of Forest Research, Yamanashi Forest Research Institute, Fujikawa, Yamanashi, Japan
[2] Laboratory of Wildlife biology, Forestry and Forest Products Research Institute, Tsukuba, Ibaraki, Japan

## ABSTRACT

Exclosures that exclude large herbivores are effective tools for the protection and restoration of grazed plant communities. However, previous studies have shown that the installation of an exclosure does not ensure plant community recovery. Our study aimed to determine the effects of the domination of unpalatable plants and the timing of exclosure installation on the plant community recovery process in montane grassland overgrazed by sika deer (*Cervus nippon*) in Japan. In this study we compared plant species composition and their cover with inside and outside exclosures installed at different times. Furthermore, we also compared them with those in 1981, when density of sika deer was very low. We used quadrats inside and outside fenced areas established in 2010 and 2011 to record both the cover and the height of species in each quadrat between 2011 and 2015. Plant cover, with the exception of graminoid species, increased in later years in all treatments. Non-metric multidimensional scaling (NMDS) plots showed significantly differentiated treatment trends. The species composition within the 2010 fenced area gradually shifted to greater similarity with the species composition reported in 1981. The plant community in the 2011 fenced area was slower to recover. Compositions of plant communities outside the fenced areas hardly changed from 2011 to 2015. Chao's dissimilarity index decreased over time between the plant community surveyed between 2011 and 2015 and the past plant community in 1981 within the exclosures, and was higher in the 2011 fenced area than in the 2010 fenced area. In conclusion, we show that the reduction of graminoids and the time after exclosure installation were important for plant community recovery from deer grazing damage. A delay in exclosure installation of one year could result in a delay in plant community recovery of more than one year.

## INTRODUCTION

Overgrazing by large herbivore populations exceeding the land-carrying capacity can reduce plant species diversity and the regeneration of tree species (*Rooney & Waller, 2003*; *Côté et al., 2004*). Fences that exclude large herbivores (exclosures) can increase the diversity of plant species within the exclosure (*Rooney, 2009*; *Yayneshet, Eik & Moe, 2009*), their

Corresponding author
Chiaki Otsu,
shiroiwayagi105@gmail.com

chances of reproduction (*Shelton & Inouye, 1995*; *Cooper, 2006*), their biomass (*Bråthen & Oksanen, 2001*; *Yayneshet, Eik & Moe, 2009*), and their heights (*Iijima & Otsu, 2018*) in some systems in several geographical regions. These previous studies may suggest that plant communities damaged by deer will recover to past compositions following exclosure installation and, as such, exclosures may be essential for the protection and restoration of plant communities.

However, in some cases, it has been noted that exclosure installation has only a limited desirable effect on forest understory vegetation restoration (*Tanentzap, Kirby & Goldberg, 2012*). Severe disturbances (i.e., high intensity, and prolonged grazing and trampling by deer) caused by the delay in the decrease or exclusion of deer may cause irreversible changes in ecosystems and shift them to "alternate (stable) states" (*Beisner, Haydon & Cuddington, 2003*; *Suding, Gross & Houseman, 2004*; *Tanentzap, Kirby & Goldberg, 2012*). More specifically, consumption of plant biomass, reduction of propagules, and formation of recalcitrant understories are suggested as the factors in the delay of plant community recovery after deer density reduction (*Tanentzap, Kirby & Goldberg, 2012*). For example, *Tamura (2016)* surveyed the plant community and soil seed bank of a forest understory within and outside exclosures after the reduction of deer density, showing that the plant community did not recover outside exclosures even though deer density had decreased. It is believed that the reason for this recovery failure was a lack of soil seed banks of deer palatable plants outside the exclosure. Therefore, early installation of exclosures may be important to avoid depletion of seed banks or propagules. In addition, the study by *Nuttle, Ristau & Royo (2014)* showed that the dominance of unpalatable plant species interfered with recovery of plant communities in a forest understory at least 20 years after deer exclusion. The dominance of unpalatable species under intense grazing is broadly observed in grasslands and forests in temperate (*Beguin, Pothier & Côté, 2011*), boreal (*Takatsuki, 2009*; *Nuttle, Ristau & Royo, 2014*), and arid zones (*Valone et al., 2002*). Therefore, the dominance of unpalatable plant species is also expected to be important for the success or failure of plant community recovery after exclosure installation.

Previous studies on the effects of exclosure installation on plant community recovery have compared species composition (*Valone et al., 2002*) and the densities of forb taxa (*Tamura, 2010*) among fenced-off areas established at different times. However, these studies examined restoration at sites that had already been disturbed by herbivores, and may therefore have overlooked the legacy of grazing effects on the plant communities. Restoration success should be measured in relation to the vegetation in reference sites at which the reconstruction effort is aimed (*SER, 2004*). The reference sites should contain primary vegetation that has not been subjected to overgrazing by large herbivores.

Many deer palatable species like forbs inhabit semi-natural grasslands (*Cousins & Lindborg, 2008*; *Otsu et al., 2017*), which have been maintained by mowing and burning. The sika deer (*Cervus nippon*) population, which has increased since the 1980s in Japan, has caused a decrease due to grazing in grassland specialist species in grassland communities (*Otsu, Hoshino & Matsuzaki, 2011*). Furthermore, the total grassland area, which had covered 50% to 70% of Japan's total land area in the 1650s, decreased to less than 1% by the early 2000s (*Ogura, 2006*) because of abandonment and conversion in land use, which

are common with European regions (*Steiner et al., 2016*). Such grassland decrease in Japan had never been occurred during past 100, 000 years (*Yamaura et al., 2019*). Therefore, the protection of grassland specialist species in semi-natural grasslands from deer herbivory has become an increasingly urgent issue (*Okubo, 2002*).

The aim of this study was to determine what factors relate to the plant community recovery process after the installation of deer exclosures in a semi-natural grassland damaged by deer. We focused on the effects of timing of exclosure installation and the dominance of unpalatable species by surveying the cover of plants in a semi-natural grassland in Japan where deer exclosures were installed, and comparing the observed species composition with the known past plant community, i.e., that hardly affected by sika deer sampled in 1981. Using our findings, we put forward recommendations for restoration projects in semi-natural grasslands.

## MATERIALS & METHODS

### Study site

The study site was at 2,003 m elevation on Mt. Kushigata (summit elevation: 2,053 m above sea level [a.s.l.]; Yamanashi Prefecture, Central Japan) in a cool-temperate zone where the mean annual precipitation and temperature (at the nearest meteorological station [Oizumi, 867 m a.s.l.]) were *ca*. 1,140 mm and 10.7 °C, respectively. The mean annual temperature at the summit of Mt. Kushigata was *ca*. 3.5 °C.

Patches (5.5 ha) of semi-natural grassland at the site were previously dominated by *Iris sanguinea* (Table S1). The patches were surrounded by *Larix kaempferi and* plantations and fragmented natural forests dominated by subalpine coniferous stands containing *Abies veitchii* and *Tsuga diversifolia*. In the past, these semi-natural grasslands were mown by the local populace (J Imakiire, pers. comm., 2010), but had been abandoned for more than 35 years at the time of this study (M Ishihara, pers. comm., 2010). This grassland is famous as a place to view various wild flowers and owned by Yamanashi prefecture (our institution). In this area, sika deer density in 2010 had been linearly increasing from $7.6/km^2$ in 2005 to $21.5/km^2$ (*Iijima & Ueno, 2016*). Thereafter, the mean density $\pm$ standard deviation of sika deer in the period 2011–2014 was $22.8 \pm 0.5/km^2$ (*Iijima & Ueno, 2016*).

### Field survey

In the summer of 1981, the Science Club at Koma High School examined the vegetation in the grassland patch we studied (*Koma High School, 1986*). The club deployed 24 precisely measured $1 \times 1$ m quadrats in a grassland typically dominated by *I. sanguinea* and sampled the vegetation inside the frames using Braun-Blanquet phytosociological procedures (*Braun-Blanquet, 1964*). The identification of species was checked by a professional researcher (E. Ohkubo). Deer distribution had not been confirmed in 1979, but was confirmed in 2003 in this area (*Biodiversty Center of Japan, 2004*), although these quadrats were not fenced at the time. Thus, we considered the species composition in 1981 to be representative of the vegetation before the beginning of heavy sika deer grazing.

In October 2010, the Minami-Alps City Government installed a $60 \times 60$ m fence, which reached a height of 2 m, to protect vegetation from deer grazing. The following year, the

Yamanashi Forest Research Institute installed a $20 \times 25$ m fence adjacent to the first fence because of heavy grazing at the site. Both fences were located on open and homogenous land with a 20 degree slope and were about 10 m from the edge of the forest, so that topographical conditions were similar for both fenced areas. We studied the vegetation in three treatments: (i) inside the fence installed in 2010 (hereafter, fence 2010); (ii) inside the fence installed in 2011 (hereafter, fence 2011); and (iii) outside both fences (hereafter, outside). We deployed 22, $1 \times 1$ m quadrats inside the exclosures (fence 2010, 10 quadrats; fence 2011, 12 quadrats), and 10 quadrats were established outside. All of the treatments were located within 20–30 m of the quadrats deployed in the Koma High School study.

We recorded the cover and the maximum height of species occurring in each of the quadrats in August of each year from 2011 to 2015 (five years total). Cover was evaluated using the Braun-Blanquet scale (*Braun-Blanquet, 1964*).

## Statistical analysis

The Koma High School study recorded only species with a cover score of $\geqq 1$ on the Braun-Blanquet scale (*Braun-Blanquet, 1964*). We adopted the same procedure for this analysis to enable good comparison between past and present conditions. Responses to grazing differ among life form groups (*Sternberg et al., 2000*; *Dupré & Diekmann, 2001*). In particular, graminoids were suggested to be tolerant to grazing in many studies (*Pellerin, Huot & Côté, 2006*; *Mysterud, 2006*; *Rooney, 2009*). Accordingly, we categorized species identified in the quadrats into five life form groups: graminoids; monocots other than graminoids (hereafter, monocots); dicot herbs (hereafter, dicots); ferns; and woody plants.

Firstly, we used a cumulative link mixed model (CLMM) with a logit link function to examine the effects of treatments (i.e., fence 2010, fence 2011, and outside), the year after exclosure installation (hereafter, the protected year), the life form of each species, and the interaction between the life form of each species and the protected year on species cover. The CLMM included each site and each year as random effects. The coefficients of the treatment and the life form were estimated when the coefficients of fence 2010 and graminoid were set as 0. The CLMM was performed by the "ordinal" package (*Christensen, 2018*) of software R (*R Core Team, 2018*).

Next, we examined the differences in plant community composition among treatments. We calculated the Chao dissimilarity index (*Chao et al., 2005*) to compare species composition in 1981 with data for the years from 2011 to 2015 across the combination of quadrats in each of the three areas. The index was used as a measure of the degree of species composition restoration. The Chao dissimilarity index was designed to consider unseen shared species using (replicated) abundance-based sampling data (*Chao et al., 2005*) and was therefore considered appropriate for our datasets, which excluded species with a cover score of <1 on the Braun-Blanquet scale, assumed as unseen rare species. We examined the ordination diagrams to evaluate species composition in 1981 and from 2011 to 2015 by non-metric multidimensional scaling (NMDS) using the Chao dissimilarity index to compare changes in species composition between treatments. NMDS was performed using the "vegan" package (*Oksanen et al., 2018*) of software R (*R Core Team, 2018*).

We examined the effects of the treatments, the protected year, and the mean cover of graminoids on the Chao dissimilarity index by a generalized linear mixed model (GLMM) with beta error distribution and a logit link function. The GLMM included each site and each year as random effects. The coefficient of treatment was estimated when the coefficient of fence 2010 was set as 0. The GLMM was performed by the "glmmTMB" package (*Brooks et al., 2017*) of software R (*R Core Team, 2018*).

For the CLMM and GLMM, we calculated the Akaike information criterion (AIC) with all possible combinations of explanatory variables. We considered the model with the lowest AIC as the most predictive model and with the explanatory variable(s) with significant effect(s) on the response variable.

## RESULTS

### The past and recent plant community

We found 56 plant species (37 dicots, 11 graminoids, 3 monocots, 2 ferns, and 3 woody plants) in all quadrats (Table S1). The total number of species identified in 1981 was 24 (Table S1), and the plant community was characterized by the dominance of a monocot (*Iris sanguinea*) and of dicots (*Serratula coronata* var *insularis*, *Geranium eriostemon* var *reinii*, *Veronicastrum sibiricum*, and *Senecio cannabifolius*). In particular, the monocot *Iris sanguinea* was highly dominant (Table S1).

The mean cover of each life form group differed among treatments and years (Fig. 1). Generally, in 2011 graminoids dominated regardless of treatments, and the plant community in 2011 differed greatly from the plant community observed in 1981. Beginning in 2012, monocots, ferns, and woody plants emerged in fence 2010. Monocots and woody plants also emerged in fence 2011 from 2014 onward. In contrast, quadrats of the outside treatment were dominated by graminoids and dicots, and no other life form species was noted during the study period. CLMM about the cover of plants with the lowest AIC included the protected year, the life form, and the interaction between the protected year and the life form (Table 1, Table S2). The results suggest that the effect of treatment on cover was not significant. Graminoid cover was the highest among the life form groups (Table 1). Plant cover of all life form types, except graminoids, increased within the exclosures over time. The coefficients of the interaction term between the protected year and the life form were positive with the exception of graminoids (Table 1).

### Changes in species composition and community structure

NMDS showed remarkably different trends in plant community composition among treatments (Fig. 2). Species composition in fence 2010 gradually became more similar to composition recorded in 1981 (Fig. 2). The species composition in fence 2011 also became similar to the composition of 1981, but recovery was slower in fence 2011 than in fence 2010 (Fig. 2). The species composition in the outside treatment remained mostly unchanged from 2011 to 2015 (Fig. 2).

The trend of plant community change was reflected in the dissimilarity indices. Dissimilarity indices of all treatments were high in 2011 (Fig. 3). However, the dissimilarity index of fence 2010 tended to decrease each year, while the dissimilarity index of fence

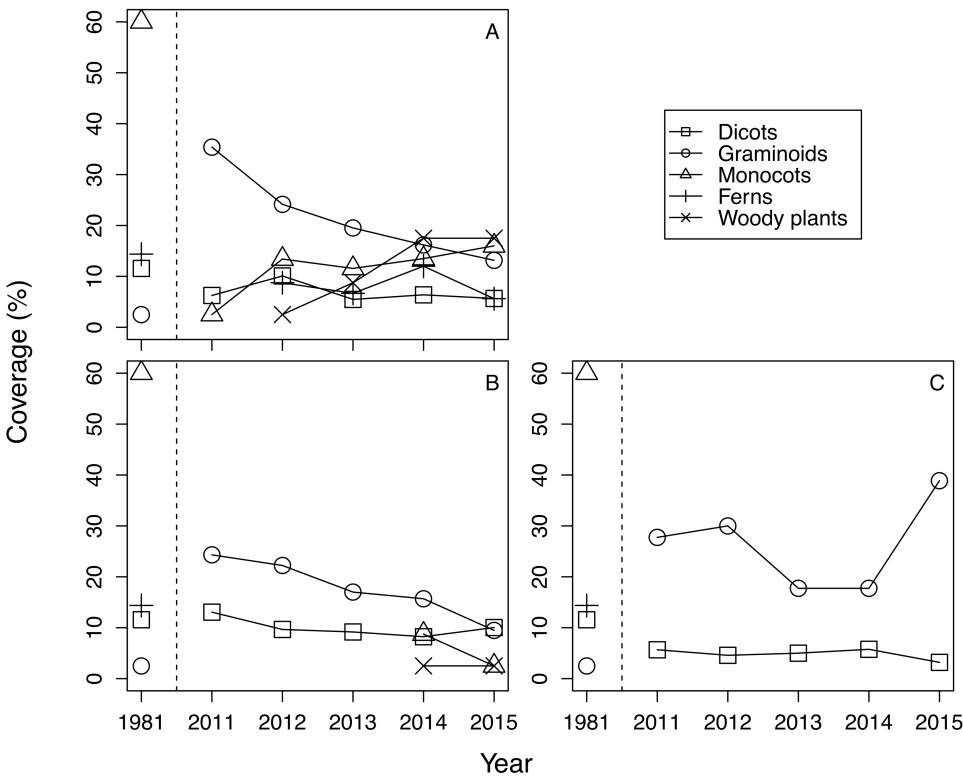

**Figure 1** **The change of mean coverage of each life form group in each treatment and year.** Species with a coverage score of ≥1 on the Braun-Blanquet scale were excluded from calculation. Covers were calculated by converting the Braun-Blanquet scores (+, 1, 2, 3, 4 and 5) into percentage cover values (0.5%, 5.5%, 17.5%, 37.5%, 62.5%, and 87.5%, respectively). (A) Fence 2010, (B) Fence 2011, (C) Outside.

2011 remained unchanged at a high level until 2013 and then decreased after 2014. In 2014, monocots and woody plants recovered in fence 2011 (Table 1, Fig. 1). The indices computed from samples outside the exclosures rarely changed during the study period (Fig. 3). The GLMM of the dissimilarity index with the lowest AIC contained all explanatory variables (Table S3). The dissimilarity index was lower in fence 2010 than in both fence 2011 and outside (Table 2). Furthermore, the dissimilarity index decreased over time with protected year and a decrease in mean cover of graminoids (Table 2).

Forb species (i.e., dicots and monocots) were not as tall as the graminoids in many quadrats regardless of the treatments in 2011 (Fig. 4), and forb species in outside remained shorter than the graminoids in 2015 (Fig. 4). However, in most of the quadrats of fences 2010 and 2011, the heights of the forb species exceeded those of the graminoids in 2015 (Fig. 4).

# DISCUSSION

**Table 1 Summary of CLMM with the lowest AIC.**

| | Estimated coefficients | | | |
|---|---|---|---|---|
| | Mean | SE[a] | z value | p value |
| The protected year | 0.329 | 0.032 | 10.175 | 0.000 |
| Life form[b] | | | | |
| Graminoids | 1.591 | 0.127 | 12.538 | 0.000 |
| Monocots | −0.930 | 0.407 | −2.285 | 0.022 |
| Ferns | −2.422 | 0.883 | −2.744 | 0.006 |
| Woody plants | −2.490 | 0.775 | −3.213 | 0.001 |
| Interaction term | | | | |
| The protected year:Graminoids | −0.362 | 0.048 | −7.466 | 0.000 |
| The protected year:Monocots | 0.417 | 0.116 | 3.603 | 0.000 |
| The protected year:Ferns | 0.547 | 0.227 | 2.411 | 0.016 |
| The protected year:Woody plants | 0.492 | 0.202 | 2.431 | 0.015 |

Notes.
[a] SE is standard error of estimated coefficients.
[b] The coefficients of the life form was estimated when the category of graminoid was set as reference category.
In this table, only final model after model selection based on AIC is shown. Please see Table S2 for AIC of models with all possible combinations of explanatory variables.

**Table 2 Summary of GLMM with the lowest AIC.**

| | Estimated coefficient | | | |
|---|---|---|---|---|
| | Mean | SE[a] | z value | p value |
| Treatments[b] | | | | |
| Fence 2011 | 1.268 | 0.135 | 9.398 | 0.000 |
| Outside | 0.749 | 0.157 | 4.772 | 0.000 |
| The protected year | −0.376 | 0.026 | −14.466 | 0.000 |
| Coverage of graminoids | 0.009 | 0.001 | 6.434 | 0.000 |

Notes.
[a] SE is standard error of estimated coefficients.
[b] The coefficient of treatment was estimated when the category of fence 2010 was set as reference category.
In this table, only final model after model selection based on AIC is shown. Please see Table S3 for AIC of models with all possible combinations of explanatory variables.

## Different response of cover to exclosure installation among life form groups

Exclusion of large herbivores from areas can alter plant cover (e.g., decrease the abundance of graminoids and increase the abundance of forbs (*Austrheim et al., 2008*; *Rooney, 2009*). In this study, we also found that plant cover of all life form groups except graminoids increased with time after exclosure (Table 1). The domination of graminoids outside the exclosure (Fig. 1) reflects the tolerance of graminoids to deer grazing. The short stature, intercalary meristems, high shoot densities, and capacity for compensatory growth enable graminoids to tolerate herbivory, giving them a competitive advantage over forb species

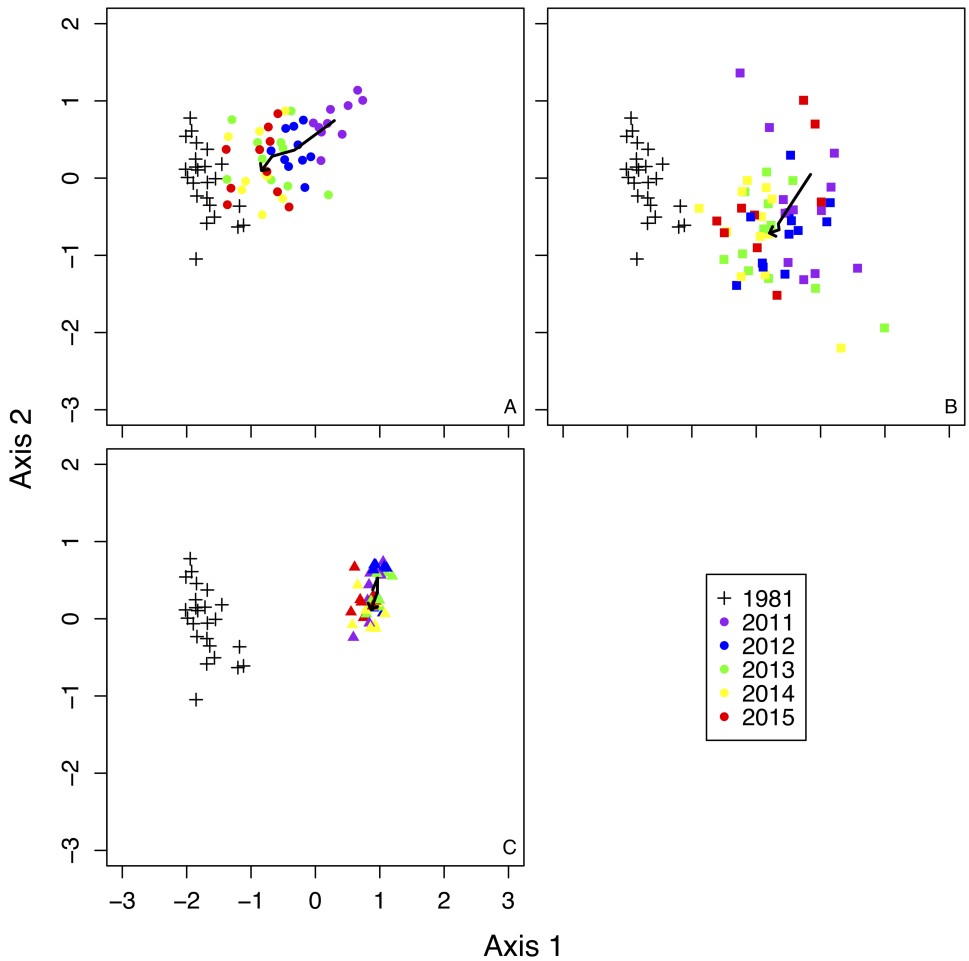

**Figure 2** Non-metric Multi-dimensional Scaling (NMDS) ordination of all sample plots in a two-dimensional space. Bold arrows in each figure indicate the changes of mean scores of NMDS by each treatment from 2011 to 2015. (A) 1981 and fence 2010, (B) 1981 and fence 2011, (C) 1981 and outside.

in heavily grazed environments (*Rooney, 2009*; *Iijima & Otsu, 2018*). Then, when grazing pressure is removed, forb species can regenerate from underground organs or seeds that are scattered from the surrounding habitat.

It should be noted that some forb species also existed and increased in cover in the outside (Table S1) where plants are exposed to deer grazing. Protection by abundant graminoids may be one of the factors that some forb species could survive in the outside. It is known that unpalatable or grazer tolerant species cover and protect neighbouring palatable species from herbivores (*Milchunas & Noy-Meir, 2002*; *Callaway et al., 2005*). Graminoids are very tolerant to grazing (*Pellerin, Huot & Côté, 2006*; *Mysterud, 2006*; *Rooney, 2009*), which could explain their high abundance in outside (Table 1). In turn, the high abundance of graminoids in the outside may have facilitated survival of forb species and their colonization outside the fences.

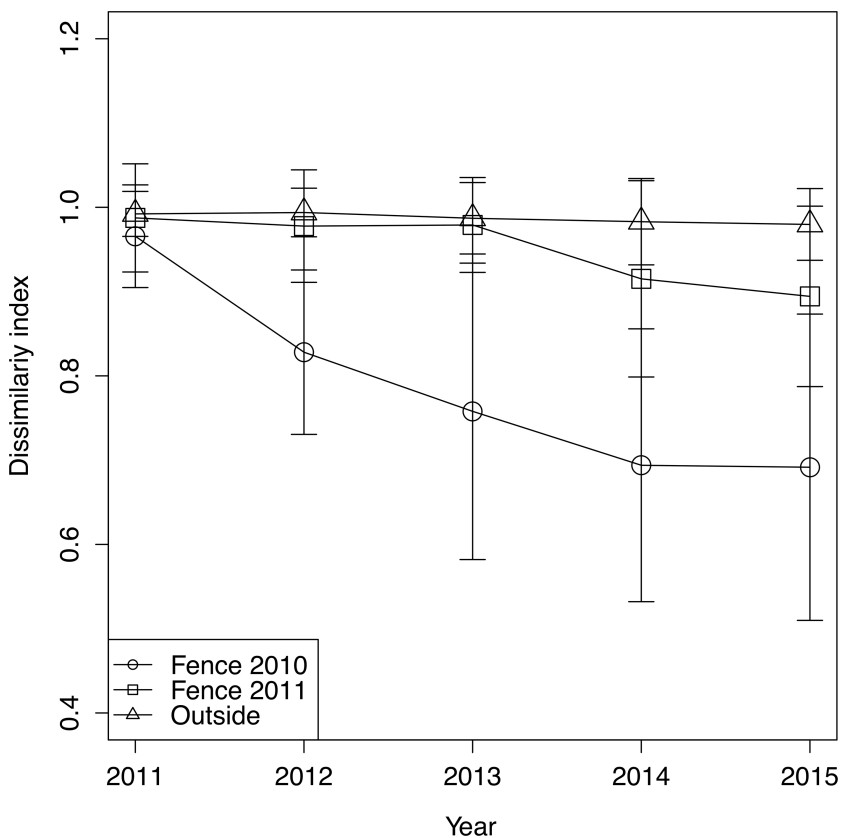

**Figure 3** **The change of dissimilarity index of each treatment and year.** Error bars indicate standard deviation of dissimilarity index.

Furthermore, some forbs may be protected from deer grazing by their diminutive size (*Bullock et al., 2001*; *Díaz, Noy-Meir & Cabido, 2001*; *Lavorel, McIntyre & Grigulis, 1999*) or rosette form (*Kahmen, Poschlod & Schreiber, 2002*; *Lavorel, McIntyre & Grigulis, 1999*; *De Villalobos & Zalba, 2010*). We found that species either not occurring or occurring at low abundance in the original vegetation (diminutive species, such as *Potentilla freyniana*, and rosette species such as *Senecio flammeus* var *glabrifolius*) were frequent in outside (Table S1).

## Determining factors for the plant community recovery process

Plant community recovery was inhibited by high cover of graminoids (Table 2). Although unpalatable plants cover palatable plants, as stated above, the facilitative effect would have been partly counterbalanced by a negative competitive interaction, whereby the graminoids would have shaded the forbs and prevented them from achieving high abundances. In fact, the heights of forbs were lower than those of graminoids in outside, while in 2015 the heights of forbs were higher than those of graminoids in enclosed areas (Fig. 4). Forb species in outside did not appear to contribute significantly to whole plant community

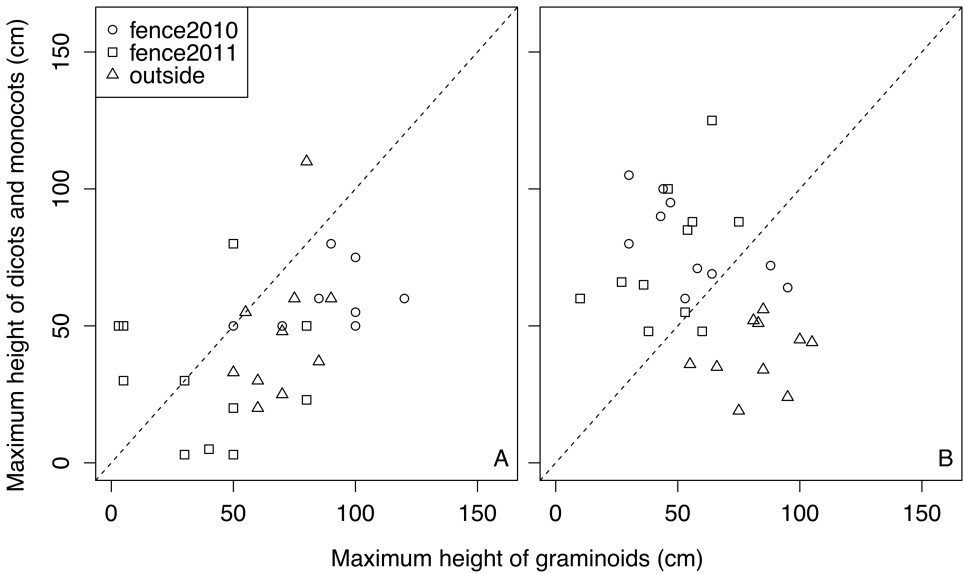

**Figure 4 Relationships between the maximum heights of graminoid species and that of forb species in each quadrat in each treatment in 2011 and 2015. A: 2011, B: 2015. .**

recovery, suggesting that the domination of unpalatable plants can delay plant community recovery.

Although the dominance of browsing-tolerant species could slow or halt plant community recovery (*Royo et al., 2010*; *Nuttle, Ristau & Royo, 2014*), in this study we observed that plant community recovery progressed over time after exclosure installation (Table 2). At the beginning of this survey there were still other plant species, such as monocots and dicots, in our research site (i.e., 2011, Fig. 1). The remaining palatable species at fence installation are suggested to be important for the successful recovery of the plant community after exclosure installation. Furthermore, although dicots did not exist in fence 2011 until 2013, dicots recovered in fence 2011 in 2014 (Fig. 1), resulting in a decreased dissimilarity index (Fig. 3).

Because we considered the effect of protected year and the random effect of each quadrat location simultaneously, early installation of exclosures may also be important to plant community recovery. The restoration of monocots including *I. sanguinea*, but excluding graminoids, seemed to be largely precluded by a one year delay in exclosure installation (Fig. 1; Table S1). Forbs (including monocots) are relatively palatable to sika deer (*Takatsuki, 1986*), and geophytes—such as members of Liliaceae, Orchidaceae, and Iridaceae—are vulnerable to grazing (*Dupré & Diekmann, 2001*; *Dorrough & Scroggie, 2008*; *Fernández-Lugo et al., 2013*). In fact, when we set fence 2011, we observed heavy grazing at the site where the fence was going to be set (C Otsu, pers. obs., 2011). Thus, both sika deer preferences for monocots other than graminoids and the poor anti-grazer defences of these plants may make their recovery difficult. At our study site, delays in the restoration of *I. sanguinea* (the most abundant species before the increase of the sika deer population)

may also have caused delays in the restoration of overall community species composition (*Nagaike, Ohkubo & Hirose, 2014*).

## CONCLUSIONS

Our study showed that plant community recovery from deer grazing damage depends on the reduction of graminoids and the time after exclosure installation in a semi-natural grassland. Specifically, a delay in exclosure installation of one year could result in a delay in plant community recovery of more than one year. Therefore, early fence installation is recommended for plant community recovery.

### Funding

The authors received no funding for this work.

### Competing Interests

The authors declare there are no competing interests.

### Author Contributions

- Chiaki Otsu conceived and designed the experiments, performed the experiments, analyzed the data, contributed reagents/materials/analysis tools, prepared figures and/or tables, authored or reviewed drafts of the paper, approved the final draft.
- Hayato Iijima performed the experiments, analyzed the data, contributed reagents/materials/analysis tools, prepared figures and/or tables, authored or reviewed drafts of the paper, approved the final draft.
- Takuo Nagaike conceived and designed the experiments, performed the experiments, contributed reagents/materials/analysis tools, authored or reviewed drafts of the paper, approved the final draft.

### Data Availability

The raw data and code are available as Supplemental Files.

### Supplemental Information

Supplemental information for this article can be found online at http://dx.doi.org/10.7717/peerj.7833#supplemental-information.

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
