# Peer review of "Plant community recovery from intense deer grazing depends on reduction of graminoids and the time after exclosure installation in a semi-natural grassland"

_PeerJ, doi:10.7717/peerj.7833_

## Round 0.1 · original submission · Major Revisions

Please take care to explain the relationship between this study and your previous work on this dataset.

Reviewer 1 ·

Basic reporting

no comment

Experimental design

no comment

Validity of the findings

no comment

Additional comments

General comments:

Below you find the review of the paper. However, during literature search I found the following publication: (Nagaike, Ohkubo & Hirose, 2014). This publication uses a survey from 2008 until 2012 and the 1981 plots as reference. This study has one author in common and was not even mentioned in the current submission. Why was this paper and its findings not mentioned given that it is dealing with the same question? I don’t think it is publishing the same data here, however I find this very suspicious.


The study has a sound data set and ask what factor might hinder a plant community to recover within a deer exclosure. Very nice is the fact that data from 1981 of the plant community is available and it is assumed these plots have not experienced very high deer grazing pressure. Besides many specific comments I have some major recommendations for the ms. First, I would recommend to include the fact that there was a full exclosure while 1981 was not a full exclosure in the discussion on why there were differences between 1981 and the recent survey. 1981 was not a exclosure meaning there was still deer herbivory, right? Second, and that’s my major criticism, the study should not aim at determining the factor that may have hindered the community to “reach” the 1981 status. It’s only the observational data of the above ground plant community available, and in order to determine what drives the system data on at least abiotic (eg.:nutrients) and/or maybe biotic (eg.:root competition, seed dispersal) and/or their interaction (eg.:plant-soil feedback) is needed. The importance of investigating such aspects has been shown for grasslands (Haynes et al., 2014) as well as forest understory (Stephan et al., 2017). The study is citing examples from grasslands and forest understory quite generally but I think it could be an advantage to better highlight in which system the dominance of a species was observed (eg.Nuttle 2014 is cited as example of a dominant unpalatable plant).
Although stated that the ms had a language check several paragraphs need a language check.


Specific comments:

1: so which factor determine the recovery? It’s better to state the main message of your study

17: is the separation in Background, Methods conform with the journal format?

18: Language: “exclosures that exclude”

20: Language: “However, previous studies shown that the installation of exclosure does not promise the plant community recovery“; from here after may only shortly indicate language issues with an “L”; I strongly recommend a professional language check

30: L increase of year

34: L in outside

50/64: yes, the effect of deer exclosure may not be the same in every case, please elaborate on this point

77: “which has increased since the 1980s” in Japan
79: “most of the semi-natural grasslands in Japan decreased drastically”, what decreased? Their abundance/area/quality…

86: from:” the factors for the plant community recovery” to: “the factors that drive the plant community recovery”
89: from:” coverage of plant” to: “coverage of plants”
91: I would use “recommendations”, rather than “guide”; also: I have not find a guide in the discussion, in fact I have not found recommendations (or maybe they have to more clearly identified. Like: “in cases… we recommend.. and it should be considered that…”)
102: you mean Table S1; “kaempferiand”
132: what does the “[10 outside]” mean? Please rephrase how many outside each; I still don’t know if it is 10 or 22 inside a fence; do you mean 2010: only inside and 2011 inside and outside?? This is very unclear

144: tolerant to…
182: “total number of species in past (i.e., 1981)” defining “past” and use it as name is not very elegant; why not just use the years as names (or “1981 plots”; “2010 plots”,…)
188: figure 1 label: change “species” to “life form”
188: please provide more details in the legend: is only the mean of the original values from all plots shown? (I guess after removal of occurrences below 1??)
191-192 and 192-194: L: please rewrite with professional language help
Table 1 and 2: I think it is not enough to show only the estimates of the minimal adequate model after backwards model selection; I prefer to see what was included from the beginning meaning either you show the starting model in the appendix or show the dropped parameters somehow; you should definitely report p-values (and the AIC itself if starting/intermediate models are shown); “Estimated coefficients of GLMM about dissimilarity index with the lowest AIC” is very uninformative; at least “Mean and standard error of estimated coefficients of a GLMM on the dissimilarity index. Only final model after model selection based on AIC is shown”
I am also confused with “when the coefficients of graminoid was set as 0”. You are playing around with the reference level of the model to get the estimates of the remaining levels? Please explain how and why this is done (I don’t think things were set as 0, but that you use Graminoids as reverence level). Alternatively, I recommend to use the Anova function from the car package in order to get the summary of your model and the emmeans and mulcomp package (glht) in order to compare estimates (for the clmm apparently car::Anova is working, see e.g.: http://rcompanion.org/handbook/G_12.html; for emmeans/ multcomp::glht you may have to check).
194: avoid using species if you have analyzed life forms
201: it is unclear what the “+” means in the figure
208: L: “and that decreased after 2014”; it is unclear if there is a decrease as there are not statistical pairwise comparisons among the estimates (please compute them using emmeans, glht and you can get the letter display using multcomp::cld); given the large sd I doubt at the moment that fence 2010 is decreasing
209: you mean table 2? And fig. 3????

227: are you sure it is resistance to grazing that makes graminoids more successful? I think it is tolerance

233: L
238: now tolerance is cited; please be careful in the hole manuscript regarding which of these fundamental different defense strategies you are talking about

238: L: tolerant to/against grazing

241: please explain; why should colonization be facilitated; I think it more likely they persisted or grown form seedbanks; one thing that I could imagine is that graminoid roots force herbal plants to grow deeper meaning they are less prone to droughts; but again, I don’t see the need to explain why there are some herbal plants left outside

247: I would not say escape grazing; there will still be some; I would rather say may experience less grazing

250: L: “These species contribute the presence”… frequently there is a “to” missing
251: are you trying to say these forbs did not majorly contributed to the dissimilarity?? Please explain
253: after having read the paper I doubt it is possible here to determine the factor explaining the recovery process; if belowground patterns, plant-soil feedbacks, nutrient cycling would have been monitored there would have been a chance to say something about what is more important to determine the observed communities

258: I am not convinced that things would counterbalance; I think it depends on many factors

261: soil compaction in dear presence is assumed to be higher as well as nutrient intake and different life forms respond differently to this; I think concluding that competitive exclusion is hindering recovery is too speculative

271: L

279: I think that hints on a major problem of your study: 1981 plots experience some disturbance (grazing, fire?) but your plots did not had any disturbance at all; please discuss this


References

Haynes AG, Schütz M, Buchmann N, Page-Dumroese DS, Busse MD, Risch AC. 2014. Linkages between grazing history and herbivore exclusion on decomposition rates in mineral soils of subalpine grasslands. Plant and Soil 374:579–591. DOI: 10.1007/s11104-013-1905-8.
Nagaike T, Ohkubo E, Hirose K. 2014. Vegetation Recovery in Response to the Exclusion of Grazing by Sika Deer ( Cervus nippon ) in Seminatural Grassland on Mt. Kushigata, Japan. ISRN Biodiversity 2014:1–6. DOI: 10.1155/2014/493495.
Stephan JG, Pourazari F, Tattersdill K, Kobayashi T, Nishizawa K, De Long JR. 2017. Long-term deer exclosure alters soil properties, plant traits, understory plant community and insect herbivory, but not the functional relationships among them. Oecologia 184:685–699. DOI: 10.1007/s00442-017-3895-3.

Reviewer 2 ·

Basic reporting

In this manuscript, the precious data gained by the high school science club in 1981 was utilized effectively, which enables to describe the effect of recent over-grazing by deer on the species compositions straightforwardly, and to indicate the degree of plant community recovery after deer exclusion fence installation by the direct comparison between past and recent data. It is worth publishing after some corrections listed below.

Experimental design

See "General comments" below.

Validity of the findings

See "General comments" below.

Additional comments

1. I have a concern about the way of use the word "original" because I don't think the situation in 1981 was "original" in this region. The authors should make it clear what is original and add some references, or change it to another word like "past" instead throughout the manuscript.

2. It is interesting that a delay in exclosure installation of only one year could result in a delay in plant community recovery of more than one year. Even though the fact may supply an important suggestion for acceleration of exclosure installation at the heavily grazed region, its reason is not discussed well. The authors partly explained the reason in L272-273 by unpublished data, but should show the data explicitly to discuss how the heavy grazing affect the plant community outside the fence 2010 and why the recovery after the exclosure installation has delayed within the fence 2011, from a plant ecological aspect. This part is especially important in this manuscript. If the authors fail to discuss in this revision, the manuscript must be rejected in the PeerJ.

3. "Conclusions" part is not informative. Add some brief explanations how these factors (such as, the dominance of graminoids and the time after the fence installation, L283) affected in the studied grassland, and make to correspond to the title of this manuscript.

Specific comments
throughout Introduction; Heavy use of "however" in the main text seems to make the logical structure poor. Reconsider the way of using conjunctions and polish up the sentences.
L107; omit either of "in this area".
L108, 110; omit "calculated by".
L120; I think Takatsuki (2009) showed the overall trend of Japanese deer population in Japan. Does this reference include the authors' studied area?
L124-127; "Although ~ construction." Unnecessary information. Omit it.
L131-132; Change to "We deployed 22 1 × 1 m quadrats inside each of the exclosure areas (fence 2010, 10 quadrats; fence 2011, 12 quadrats), and 10 outside."
L134-135: "vegetation ~ grazing." Omit.
L151 cumulative "logit" mixed model?

Discussion
L274-279; These sentences are too descriptive and unnecessary. They don't include anything found from this study.

Figure2; What does "+" indicate?
Table1, Table2; Show the z-value, the probability (P value) and the statistical significance in each parameter.

---

## Round 0.2 · Minor Revisions

Thank you for your thorough responses to the reviewer comments. I have marked up your manuscript with some suggested minor changes that I hope you will find helpful. Unfortunately I am only able to send my suggestions as a PDF file, so please contact me directly by email if you would prefer to get the Word version.

---

## Round 0.3 · accepted · Accept

Thank you for responding rapidly to the revisions.